# The *Arabidopsis* Ca^2+^-Dependent Protein Kinase CPK12 Is Involved in Plant Response to Salt Stress

**DOI:** 10.3390/ijms19124062

**Published:** 2018-12-14

**Authors:** Huilong Zhang, Yinan Zhang, Chen Deng, Shurong Deng, Nianfei Li, Chenjing Zhao, Rui Zhao, Shan Liang, Shaoliang Chen

**Affiliations:** 1Beijing Advanced Innovation Center for Tree Breeding by Molecular Design, College of Biological Sciences and Technology, Beijing Forestry University, Beijing 100083, China; hlzhang2018@126.com (H.Z.); xhzyn007@163.com (Y.Z.); ced501@163.com (C.D.); danceon@126.com (S.D.); nl1669@nyu.edu (N.L.); 1120170396@mail.nankai.edu.cn (C.Z.); lschen@bjfu.edu.cn (S.C.); 2Beijing Advanced Innovation Center for Food Nutrition and Human Health, School of Food and Chemical Engineering, Beijing Technology and Business University, Beijing 100048, China

**Keywords:** *Arabidopsis*, CDPK, ion homeostasis, NMT, ROS, salt stress

## Abstract

CDPKs (Ca^2+^-Dependent Protein Kinases) are very important regulators in plant response to abiotic stress. The molecular regulatory mechanism of CDPKs involved in salt stress tolerance remains unclear, although some CDPKs have been identified in salt-stress signaling. Here, we investigated the function of an *Arabidopsis* CDPK, CPK12, in salt-stress signaling. The *CPK12*-RNA interference (RNAi) mutant was much more sensitive to salt stress than the wild-type plant GL1 in terms of seedling growth. Under NaCl treatment, Na^+^ levels in the roots of *CPK12*-RNAi plants increased and were higher than levels in GL1 plants. In addition, the level of salt-elicited H_2_O_2_ production was higher in *CPK12*-RNAi mutants than in wild-type GL1 plants after NaCl treatment. Collectively, our results suggest that CPK12 is required for plant adaptation to salt stress.

## 1. Introduction

Saline soil cannot be used for agriculture and forestry production [1], and soil salinity is a major abiotic stress for plants worldwide [2,3]. When plants suffer from salt environments, the accumulation of sodium and chloride ions breaks the ion balance and causes secondary stress, such as oxidative bursts [4,5].

Plants have evolved sophisticated regulatory mechanisms to avoid and acclimate to salt stress and repair related damage, processes based on morphological, physiological, biochemical and molecular changes [6]. Salt overly sensitive (SOS) signaling is the most important pathway for regulating plant adaptation to salt stress [4,7]. In *Arabidopsis*, salt-induced increases in cytoplasmic calcium (Ca^2+^) are sensed by the EF-hand–type Ca^2+^-binding protein SOS3. Ca^2+^ together with SOS3 activates SOS2, a serine/threonine protein kinase. Activated SOS2 phosphorylates and stimulates the activity of SOS1, a plasma membrane–localized Na^+^/H^+^ antiporter, leading to regulation of ion homeostasis during salt stress [8,9,10,11]. A Na^+^/H^+^ exchanger, which is localized to plasma membrane, also plays an important role in *Populus euphratica*, the roots of which exhibit a strong capacity to extrude Na^+^ under salt stress; furthermore, the protoplasts from root display enhanced Na^+^/H^+^ transport activity [12]. In addition, wheat *Nax1* and *Nax2* affect activity and expression levels of the SOS1-like Na^+^/H^+^ exchanger in both root cortical and stellar tissues [13].

Salt stress increases the production of reactive oxygen species (ROS), which plays a dual role in plants: they function as toxic byproducts of metabolism and as important signal transduction molecules [14,15,16]. Peroxisomes and chloroplasts are the major organelles of ROS generation [17,18,19], and plants eliminate ROS through non-enzymatic and enzymatic scavenging mechanisms [20]. Non-enzymatic antioxidants include the major cellular redox buffers glutathione and ascorbate, as well as flavonoids, carotenoids, alkaloids, and tocopherol [21]. Enzymatic ROS scavenging pathways in plants include superoxide dismutase (SOD), ascorbate peroxidase (APX), glutathione peroxidase (GPX), and catalase (CAT) [20]. H_2_O_2_ is the end-product of SOD, which is harmful to DNA, proteins, and lipids [20]. Halophytes can send stress signals quickly through H_2_O_2_, and have an efficient antioxidant ability to scavenge H_2_O_2_ upon completion of signaling [22]. In addition, H_2_O_2_ is a signaling molecule in the plant response to salt stress [23,24]. *P. euphratica* responds to salt stress with rapid H_2_O_2_ production, and exogenous H_2_O_2_ application enhances the Na^+^/H^+^ exchange [25]. Pharmacological experiments have strongly indicated that NaCl-induced Na^+^/H^+^ antiport is inhibited when H_2_O_2_ is absent [25], and H_2_O_2_-regulated K^+^/Na^+^ homeostasis in the salt-stressed plant is Ca^2+^-dependent [25]. Exogenous H_2_O_2_ causes elevated cytosolic Ca^2+^ [25], which stimulates plasma membrane–localized Na^+^/H^+^ antiporters through the SOS signaling pathway [2,4,5]. Furthermore, H_2_O_2_ mediates increased *SOS1* mRNA stability in *Arabidopsis* and may therefore contribute to cellular Na^+^ protection [26].

Ca^2+^ is a conserved second messenger in plant growth and development pathways and contributes to plant adaptations to environmental challenges [27,28]. In plants, calmodulin (CAM), calcineurin B-like proteins (CBL), and Ca^2+^-dependent protein kinases (CDPKs) are important Ca^2+^ sensors [29,30,31,32,33]. For CDPKs, *Arabidopsis* has 34 members, rice (*Oryza sativa*) has 29 members, wheat (*Triticum aestivum*) has 20 members, and poplar (*Populus trichocarpa*) has 30 members [34,35,36,37]. In recent years, CDPKs have been characterized as playing an important function in mediating stress-signaling networks [38,39,40].

Genetic and biochemical evidence implicates several CDPKs in plant adaptations to environmental stress. *Arabidopsis* CPK32 (Ca^2+^-Dependent Protein Kinase 32) phosphorylates ABF4 (ABRE Binding Factor 4) to participate in abscisic acid (ABA) signaling [41]. CPK4 and CPK11 are important positive regulators mediating ABA signaling pathways [42], but their homolog, CPK12, plays a negative role in this signaling [43,44]. CPK10 interacts with HSP1, which contributes to plant drought responses by modulating signaling through ABA and Ca^2+^ [45]. CPK23 responds to drought and salt stresses, and together with CPK21 constitutes a pair of critical Ca^2+^-dependent regulators of the guard cell anion channel SLAC1 (Slow Anion Channel-Associated 1) in ABA signaling [46,47,48]. CPK3 and CPK6 positively regulate ABA signaling in stomatal movement [49,50,51], and CPK6 functions as a positive regulator of methyl jasmonate signaling in guard cells [52]. CPK13 inhibits opening of the stomata through its inhibition of guard cell–expressed KAT2 (K^+^ transporter 2) and KAT1 (K^+^ transporter 1) channels [53]. The expression of *CPK27* is induced by NaCl, and the *cpk27-1* mutant is much more sensitive to salt stress than wild-type plants in terms of seed germination and post-germination seedling growth [54].

Overexpression of rice *CPK21* enhances rice’s capacity to tolerate high salinity [55]. OsCPK12 reduces ROS levels to regulate salt tolerance [56] and plays a positive role in plant responses to drought, osmotic stress, and dehydration [57]. In maize (*Zea mays*), ZmCCaMK is required for ABA-induced antioxidant defense, and the ABA-induced activation of ZmCCaMK is required for H_2_O_2_-dependent nitric oxide production [58]. ZmCPK4 positively regulates ABA signaling and enhances drought stress tolerance in *Arabidopsis* [59], while ZmCPK11 functions upstream of ZmMPK5 and regulates ABA-induced antioxidant defense [60]. The expression of *PeCPK10*, a gene cloned from *P. euphratica*, is induced by salt, drought and cold treatments, overexpression of *PeCPK10* in *Arabidopsis* improves the plant’s tolerance of freezing [61]. In grape berry, ABA stimulates ACPK1 (ABA-stimulated calcium-dependent protein kinase1), which is potentially involved in ABA signaling [62]. Heterologous overexpression of ACPK1 in *Arabidopsis* promotes significant plant growth and enhances ABA sensitivity in seed germination, early seedling growth, and stomatal movement, providing evidence that ACPK1 is involved in ABA signal transduction as a positive regulator [63].

Although the functions of CDPKs in plant response to environmental stress have been demonstrated, the molecular biological mechanisms of CDPKs remain unclear. Previously, we reported that *Arabidopsis* CPK12 negatively regulates ABA signaling [43,44]. Here, we show that CPK12 mediates salt stress tolerance by regulating ion homeostasis and H_2_O_2_ production. Down-regulation of *CPK12* results in salt hypersensitivity in seedling growth and accumulation of higher levels of Na^+^ and H_2_O_2_. Our results show that CPK12 may modulate salt stress tolerance in *Arabidopsis*.

## 2. Results

### 2.1. Identification of RNA Interference (RNAi) Mutants of CPK12

We previously identified the function of *Arabidopsis CPK12*, generated *CPK12*-RNAi lines, and observed that down-regulation of *CPK12* results in ABA hypersensitivity in seed germination and post-germination growth. CPK12 interacted with and phosphorylated and stimulated the type 2C protein phosphatase ABI2. In addition, CPK12 together with ABI2 negatively regulates ABA signal transduction [43]. Thus, we wondered whether *CPK12* was involved in salt-stress signaling. To address this question, we re-generated *CPK12*-RNAi lines and selected four (lines *R1*, *R4*, *R7*, *R8*) as examples for this study. Expression of *CPK12* was down-regulated in these RNAi lines, and the level of *CPK12* mRNA gradually decreased from line *R8* to line *R1*, creating a gradient of *CPK12* expression levels (Figure 1). In addition, the expression of a control gene *EF-1α* (*Elongation Factor-1α*), which is not related to salt-stress, was not affected in those *CPK12*-RNAi lines (Figure 1).

### 2.2. Down-Regulation of CPK12 Results in NaCl Hypersensitivity in Seedling Growth

We next examined whether CPK12 affects seedling growth under salt stress. Seeds from the *CPK12*-RNAi mutant and GL1 plants were sown on medium containing various concentrations of NaCl. The presence of 110–150 mM NaCl inhibited *CPK12*-RNAi mutant growth. In addition, the cotyledons of the *CPK12*-RNAi mutants were chlorotic compared with GL1 seedlings (Figure 2).

*CPK12*-RNAi mutants exhibited the same post-germination seedling growth status as GL1 plants in the free of NaCl medium; however, compared with GL1 plants, NaCl suppressed root growth of *CPK12*-RNAi mutants more strongly, the reduction in the growth of salt-stressed *CPK12*-RNAi seedlings was more pronounced than for GL1 seedling (Figure 3). Taken together, these results suggest that *CPK12* is involved in salt-stress tolerance in *Arabidopsis*.

### 2.3. Salt Stress Induced the Ca^2+^ Elevation in Root Tissue

We examined the Ca^2+^ level in the *CPK12*-RNAi plants and wild type plants GL1 after salt stress using the Ca^2+^ specific probe, Rhod-2 AM. In the absence of salt treatment, the relative fluorescence intensity was not significantly different between GL1 and *CPK12*-RNAi plants, except the R1 line, probably due to the lowest expression of *CPK12* in R1 line. Under salt treatment, as expected, the Ca^2+^ levels in the roots of *CPK12*-RNAi plants and GL1 increased (Figure 4).

### 2.4. Down-Regulation of CPK12 Leads to Na^+^ Accumulation in Root Tissue

To investigate the cause of the observed hypersensitivity of *CPK12*-RNAi mutants to salt stress, sodium accumulation in root cells was examined using the sodium-specific dye CoroNa-Green. Under no salt stress, CoroNa-Green fluorescence was almost undetectable in the root cells of *CPK12*-RNAi and wild-type plants because of low Na^+^ content in root cells. Under NaCl treatment, Na^+^ levels in the roots of *CPK12*-RNAi plants increased and were higher than levels in GL1 plants (Figure 5).

To determine whether CPK12 contributed to the regulation of ion homeostasis, NMT (Non-invasive Micro-test Technique) was used to record root Na^+^ fluxes in the *CPK12*-RNAi plants and GL1 plants under a long-term NaCl treatment (0, 110, 120, 130 mM; 7 d). In the absence of salt stress, Na^+^ efflux in the apical region of roots was not significant between *CPK12*-RNAi plants and wild type GL1 plants. However, long-term salt treatment caused a rise in Na^+^ efflux in GL1 plants and *CPK12*-RNAi mutants, but this was more pronounced in GL1 plants than in *CPK12*-RNAi plants (Figure 6). These observations indicate that CPK12 participates in salt-stress tolerance by regulating ion balance in root tissue.

### 2.5. Down-Regulation of CPK12 Results in H_2_O_2_ Burst and Accumulation

ROS accumulates when plants are exposed to salt stress, so we investigated H_2_O_2_ levels in *CPK12*-RNAi plants using a H_2_O_2_-specific fluorescent probe, H_2_DCF-DA. In the NaCl shock condition, the levels of H_2_O_2_ in *CPK12*-RNAi plants were higher than GL1 plants (Figure 7). The salt stress-induced H_2_O_2_ accumulation in *Arabidopsis* was also detected after 12 h or 24 h treatment; compared with GL1 plants, the level of H_2_O_2_ in *CPK12*-RNAi plants was significantly higher after exposure to high NaCl concentrations (Figure 7).

We measured the activities of antioxidant enzymes, such as SOD, CAT, and APX, in GL1 and *CPK12*-RNAi plants. In the absence of salt treatment, the activities of SOD and CAT were not significantly different between GL1 and *CPK12*-RNAi plants, but the activity of APX in *CPK12*-RNAi plants was lower than GL1. Under salt stress conditions, the activity of SOD in GL1 was higher than *CPK12*-RNAi plants, but CAT and APX were lower in *CPK12*-RNAi plants, when compared with GL1 (Figure 8). Taken together, these data imply that CPK12 is involved in the elimination of H_2_O_2_ under salt stress.

### 2.6. Down-Regulation of CPK12 Suppressed Cell Viability in Arabidopsis Roots

Previous studies showed that a high level of NaCl could reduce viability and increase programmed cell death in plants [64,65,66,67]. Cell viability was assayed with fluorescein diacetate (FDA) to determine whether salt stress could induce cell death in *CPK12*-RNAi plants. FDA staining showed an effect of salt treatment on cell viability in the elongation zone of roots. Wild-type GL1 and *CPK12*-RNAi plants grown in control conditions (NaCl-free Murashige–Skoog (MS) medium) showed clear FDA fluorescence with the cytoplasm of root cells, features indicating that the cells were viable. However, in salt-stressed *CPK12*-RNAi plants, the FDA fluorescence was undetectable in a number of root cells, and the fluorescence intensity was reduced (Figure 9). Compared to wild-type GL1 plants, the *CPK12*-RNAi plants exhibited lower cell viability during the period of salt stress.

### 2.7. Down-Regulation of CPK12 Alters the Expression of Some Salt-Responsive Genes

We tested the expression of the following salt-related genes in the GL1 and *CPK12*-RNAi plants: *SOS1*, *SOS2*, and *SOS3* [4,7,8,9,10,11], *AHA1* and *AHA2* [68], *PER1* [69,70], *SOD*, *CAT*, and *APX*. In the absence of salt treatment, the expression of *AHA1* was not significantly difference between wild-type GL1 plants and *CPK12*-RNAi plants, but the expression of *SOS1*, *SOS2*, *SOS3 AHA2*, and *PER1* was down-regulated in *CPK12*-RNAi plants (Figure 10). Under salt stress, down-regulation of *CPK12* did not affect expression of *AHA2* and *PER1*, but significantly reduced expression of *SOS1* in *R8*, *R7*, and *R4* plants, *SOS2* in *R7*, *R4*, and *R1* plants, *AHA1* in *R8*, *R7*, *R4*, and *R1* plants (Figure 10). In the absence of salt treatment, the expression of *SOD* and *CAT* was down-regulated in *CPK12*-RNAi plants, and *APX* was down-regulated in *R8*, *R7*, and *R4* plants. Under salt stress, the expression of *APX* was down-regulated in *R1*. It is interesting that the expression of *SOD*, *CAT*, and *APX* was nearly undetectable in *R8* and *R7* plants, whether under salt or no-salt stress (Figure 10).

## 3. Discussion

### 3.1. CPK12 Is Involved in Salt Stress Tolerance in Plants

In this investigation, *Arabidopsis* CPK12 was identified and characterized as a regulatory component involved in salt tolerance in terms of seedling growth. Previously we reported that CPK12 negatively regulates ABA signal transduction [43,44]. These results together imply an important function of CPK12 in regulating plant salt stress tolerance. Of note, although the expression of some salt-related genes were down-regulated in *CPK12*-RNAi plants without salt stress treatment (except *AHA1)*, down-regulated *CPK12* did not influence post-germination seedling growth, and the expression of a control gene *EF-1α* that is not related to salt-stress, indicating that CPK12 is involved in salt stress signal transduction, but is not related to seedling development. Previously we reported that CPK27’s function in salt stress tolerance [54], although CPK12 and CPK27 had similar functions. Our results suggest that the function of CPK12 may not be redundant for CPK27, because independent down-regulation of CPK12 or CPK27 can change the responses of *Arabidopsis* seedlings to salt stress. Current evidence suggests that CDPKs regulate plant tolerance to abiotic stress through ABA and Ca^2+^ pathways. For example, CPK10 regulates plants response to drought stress through ABA- and Ca^2+^- mediated stomatal movements [45]; CPK4 and CPK11 are positive regulators in Ca^2+^-mediated ABA signaling [42]. Thus, CPK12 and other CDPK members constitute a complicated regulation network, which functions in plant adaptation to salt and drought stresses.

### 3.2. CPK12 Regulates Na^+^ Balance in Salt-Stressed Plants

The ability to retain ion balance is very important for plant survival in saline environments [1,5,71]. Here, under salt stress, wild-type plants GL1 and *CPK12*-RNAi absorbed and accumulated Na^+^ in roots; when compared with GL1, Na^+^ accumulation was significantly higher in the roots of *CPK12*-RNAi plants, and net Na^+^ efflux was reduced in *CPK12*-RNAi roots compared to GL1 plants. There are many CDPKs that can interact and regulate the activity of ion transporters. Under drought stress, AtCPK23 phosphorylates the guard cell anion channel SLAC1, which is collaborated with activation of the potassium-release channel GORK to regulate stomatal movement [47]. AtCPK3 and AtCPK6 are specifically expressed in guard-cell, regulate guard cell S-type anion channels and contribute to stomatal movement [49]. AtCPK13 phosphorylates two inward K^+^ channels, KAT1 and KAT2, to restrict the stomatal aperture, and AtCPK3 phosphorylates and activates a two pore K^+^ channel TPK1 [53,72]. Thus, like other CDPK members, it is speculated that CPK12 may regulate Na^+^ balance in salt-stressed *Arabidopsis* seedlings. Compared with previous studies wherein some CDPKs were localized in guard cells to regulate stomatal aperture, our results give a new insight and show that CPK12 may function in root systems to regulate ion balance. CDPKs therefore constitute a network, which at the whole-plants level in roots and shoots, uses interaction and phosphorylation to regulate ion transporters or channels to improve plant tolerance to drought or salt stress over the short term.

### 3.3. CPK12 Regulates ROS Homeostasis in Salt-Stressed Plants

ROS, such as the superoxide anion, accumulates in stressed plants. Plasma membrane NADPH oxidases generate superoxide anions, which are transformed into H_2_O_2_ by superoxide dismutase [20]. In this study, compared with GL1 plants, *CPK12*-RNAi plants accumulated more H_2_O_2_ in roots, irrespective of the duration of NaCl treatment. Without salt-stress, compared with GL1, the expression of *CAT* gene was down-regulated in *CPK12*-RNAi plants, but the activity of CAT was not reduced, and the level of H_2_O_2_ was not significantly different between GL1 and *CPK12*-RNAi plants. Furthermore, under salt stress treatment, the activity of SOD was higher in *CPK12*-RNAi plants than GL1, but the activity of CAT was lower in *CPK12*-RNAi plants than GL1. The enhanced activity of SOD in *CPK12*-RNAi plants results in H_2_O_2_ production, but the reduced activity of CAT lead to H_2_O_2_ accumulation; these results suggest that down-regulated *CPK12* cannot scavenge NaCl-induced H_2_O_2_ bursts. Previous studies showed that down-regulated expression of CDPKs, which are related to antioxidases, may cause the accumulation of H_2_O_2_. *Arabidopsis cpk27-1* mutants accumulate more H_2_O_2_ in roots [54]. Potato StCDPK4 and StCDPK5 regulate the production of ROS [73]. In rice, *OsCPK12*-OX plants accumulate less H_2_O_2_ under conditions of high salinity, and this accumulation is more pronounced in *oscpk12* mutants and *OsCPK12* RNAi plants [56]. Similarly, excess H_2_O_2_ leads to oxidative damage and growth inhibition in *CPK12*-RNAi plants under salinity conditions. In contrast, *Arabidopsis cpk5 cpk6 cpk11 cpk4* quadruple mutants harbor decreased ROS content, suggesting that these CDPKs regulate ROS production potentially by phosphorylating the NADPH oxidase RBOHB [74]. Therefore, CDPKs play key roles in regulating ROS production and accumulation in plants [75].

### 3.4. CPK12 Functions with Potential Substrates in Salt Stress Signaling

In recent years, many substrates of CDPKs were identified. The transcription factor ABF4 is a substrate of CPK4/11 in *Arabidopsis* [42]. CPK12 can phosphorylate type-2C protein phosphatase ABI2 [43]. HSP1 interacts with CPK10 [45], SLAC1 is an interacting partner of CPK21 and CPK23 [47], and CPK13 specifically phosphorylates KAT2 and KAT1 [53]. In nutrient signaling, CPK10, CPK30, and CPK32 could potentially phosphorylate and activate all NLPs and possibly other transcription factors with overlapping or distinct target genes to support transcriptional, metabolic, and system-wide nutrient-growth regulations [76]. Potato CDPK5 phosphorylates the N-terminal region of plasma membrane RBOH (respiratory burst oxidase homolog) protein and participates in RBOHB-mediated ROS bursts, conferring resistance to near-obligate pathogens but increasing susceptibility to necrotrophic pathogens [77]. In this work, CPK12 was involved in plant adaptation to salt stress by regulating Na^+^ and H_2_O_2_ homeostasis. These results indicate that CPK12 may interact with and phosphorylate several salt stress-related proteins as potential substrates in its regulatory function. To deeply demonstrate the regulatory mechanism of CPK12, the downstream components of CPK12 need to be identified, and their relationship with the whole complex CDPK regulation network need to be elucidated. Although the functions of CDPKs are widely identified in recent years, the complete CDPK signal transduction pathway is still not clearly illustrated, as the CDPK transduction network is very complex. Future progress is likely to identify sensors, channels, and other regulators involved in generating complex Ca^2+^ signatures in plant responses to hormones and environmental cues.

## 4. Materials and Methods

### 4.1. Plant Materials, Constructs, and Arabidopsis Transformation

*Arabidopsis thaliana* GL1 (Col-5) was used in this work for generating the *CPK12*-RNAi plants. A specific 242-bp fragment of *CPK12* (At5g23580) corresponding to the region of nt 6 to 247 was amplified with forward primer 5′-ACGCGTCGACGAACAAACCAAGAACCAGATGGGTT-3′ and reverse primer 5′-CCGCTCGAGCGTTGGGGTATTCAGACAAGTGATG-3′. This fragment was inserted into the pSK-int vector, which was digested with the *Xho*I and *Sal*I. The same fragment, amplified with forward primer 5′-AACTGCAGGAACAAACCAAGAACCAGATGGGTT-3′ and reverse primer 5′-GGACTAGTCGTTGGGGTATTCAGACAAGTGATG-3′, was inserted into the pSK-int carrying the previous fragment. The entire RNAi cassette linked with actin 11 intron was excised from pSK-int vector, and inserted into the *Sac*I *Apa*I digested vector pSUPER1300(+) [78]. The construct was introduced into *Agrobacterium tumefaciens* GV3101 and transformed into GL1 by the floral dip method [79]. Transgenic plants were grown on MS agar plates containing hygromycin (50 μg/mL) to screen for positive seedlings. The homozygous T3 seeds of the transgenic plants were used for further analysis. Plants were grown in a growth chamber at 20–21 °C on MS medium at about 80 μmoL photons m^−2^·s^−1^, or in compost soil at about 120 μmoL photons m^−2^·s^−1^ over a 16-h photoperiod.

### 4.2. qRT-PCR Analysis

To assay the gene expression in the transgenic plants, quantitative real-time PCR analysis was performed with the RNA samples isolated from 10-day-old seedlings. For *CPK12*, qRT-PCR amplification was performed with forward primer 5′-CGAAACCCTCAAAGAAATAA-3′ and reverse primer 5′-TGGTGTCCTCGTACGCACTCTC-3′. The primers specific for salt-related genes were: forward 5′-CACAAACATTTACCGAAAACCA-3′ and reverse 5′-CAAATTTGCAAAGCTCATATCG-3′ for *AHA1* (At2g18960); forward 5′-TGACTGATCTTCGATCCTCTCA-3′ and reverse 5′-GAGAATGTGCATGTGCCAAA-3′ for *AHA2* (At4g30190); forward 5′-CGTGCCCTTCATATTGTTGG-3′ and reverse 5′-GACGCCATCAACAACGAGTC-3′ for *PER1* (At1g48130); forward 5′-GTGAAGCAATCAAGCGGAAA-3′ and reverse 5′-TGCGAAGAAGGCGTAGAACA-3′ for *SOS1* (At2g01980); forward 5′-GCGAACTCAATGGGTTTTAAGT-3′ and reverse 5′-CTTACGTCTACCATGAAAAGCG-3′ for *SOS2* (At5g35410); forward 5′-CCGGTCCATGAAAAAGTCAAAT-3′ and reverse 5′-CTCTTTCAATTCTTCTCGCTCG-3′ for *SOS3* (At5g24270); forward 5′-AGGAAACATCACTGTTGGAGAT-3′ and reverse 5′-GAGTTTGGTCCAGTAAGAGGAA-3′ for *SOD* (At1g08830); forward 5′-AGGATCAAACTTTGAGGGGTAG-3′ and reverse 5′-CTTGTGGTTCCTGGAATCTACT-3′ for *CAT* (At1g20620); forward 5′-GATGTCTTTGCTAAGCAGATGG-3′ and reverse 5′-GAGTTGTCGAAGATTAGAGGGT-3′ for *APX* (At1g07890); forward 5′-CACCACTGGAGGTTTTGAGG-3′ and reverse 5′-TGGAGTATTTGGGGGTGGT-3′ for *EF-1α* (At5g60390). Amplification of *ACTIN2*/*8* (forward primer 5′-GGTAACATTGTGCTCAGTGGTGG-3′, reverse primer 5′-AACGACCTTAATCTTCATGCTGC-3′) gene was used as an internal control.

### 4.3. Phenotype Identification

For the seedling growth experiment, seeds were germinated after stratification on MS medium supplemented with various concentrations of NaCl. Seedling growth was examined 10 days after stratification.

### 4.4. Measurement of Cytosolic Ca^2+^ Concentrations

For cytosolic Ca^2+^ concentration analysis, the Ca^2+^-specific fluorescent probe Rhod-2 AM (Invitrogen, Carlsbad, CA, USA) was used to measure the concentration of Ca^2+^ as previously described [80]. Briefly, CPK12-RNAi mutants and GL1 seedlings were treated with MS liquid solution supplemented with or without 100 mM NaCl for 12 h. Then, control and salinized plants roots were 2 µM Rhod-2 AM (prepared in MS liquid solution, pH 5.8) incubated in the dark for 1 h at room temperature. Then, the Arabidopsis plants were washed four to five times with distilled water. The image of Ca^2+^ fluorescence in the probe-loaded roots was measured with a Leica SP5 confocal microscope (Leica. Microsystems GmbH, Wetzlar, Germany), with emission at 570–590 nm and excitation at 543 nm [80].

### 4.5. Detection of Cytosolic Na^+^ Concentrations

Root cellular Na^+^ levels were detected with Na^+^-specific fluorescent probe, CoroNa™ Green, AM (Invitrogen, Carlsbad, CA, USA). Seedlings of wild-type GL1 and *CPK12*-RNAi mutants were treated with 0, 120, or 150 mM NaCl in MS liquid solution for 12 h. Then, control and salinized seedlings were incubated with CoroNa in the dark for 1 h, and washed 3–4 times with distilled water subsequently. Na^+^ fluorescence was observed with a Leica SP5 confocal microscope (excitation: 488 nm; emission: 510–530 nm, Microsystems GmbH, Wetzlar, Germany) [25,80,81]. ImageJ software (Version 1.48, National Institutes of Health, Bethesda, MD, USA) was used to quantify relative fluorescence intensity. It is worth noting that there are several commercial Na^+^ specific probes; for example, SBFI, Sodium Green, CoroNa etc., Sodium Green displays a modest fluorescence increase in response to Na^+^ binding [82,83], while CoroNa is more suitable for detecting Na^+^ in a wider range of concentrations; and the selectivity of CoroNa is 4 times higher to Na^+^ than to K^+^ binding [82], but CoroNa is not suitable for the detection of relatively low Na^+^ changes in cells [83]. In this work, after NaCl treatment, the roots absorbed and accumulated high levels of Na^+^, and Na^+^ efflux was inhibited. Our previous studies showed that CoroNa is suitable for detecting Na^+^ level after NaCl treatment in tobacco, *Arabidopsis*, and *Glycyrrhiza uralensis* [84,85,86]; thus, we selected CoroNa Green to detect the cytosolic Na^+^ level in this work.

### 4.6. Net Fluxes Measurements of Na^+^

Net Na^+^ flux was measured using the NMT technique (NMT-YG-100, Younger, Amherst, Massachusetts, USA) as described previously [12,80]. One-week-old seedling grown on MS medium containing 0, 110, 120, 130 mM NaCl was washed 4–5 times with ddH_2_O and transferred to the measuring chamber containing 10–15 mL measuring solution, which included 0.1 mM NaCl, 0.5 mM KCl, 0.1 mM CaCl_2_, 0.1 mM MgCl_2_, and 2.5% sucrose, pH 5.8. After the roots were immobilized to the bottom of the chamber, Na flux measurements were started at 200–300 μm from the root apex. The Na^+^ flux was continuously recorded for 17–20 min. The Na^+^ flux was detected by shifting the ion-selective microelectrode between two sites close the roots over a preset length (30 μm for intact roots in this experiment) at a frequency in the range of 0.3–0.5 Hz. The electrode was stepped from one site to another in a predesigned sampling routine, while the sample was also scanned with a 3-D microstepper motor manipulator (CMC-4). Pre-pulled and salinized glass micropipettes (4–5 μm aperture, XYPG120-2; Xuyue Sci. and Tech. Co., Ltd., Beijing, China) were processed with a backfilling solution (Na: 250 mM NaCl) to a length of approximately 1 cm from the tip, then front-filled with about 10 μm columns of selective liquid ion-exchange cocktails (Na: Fluka 71178). An Ag/AgCl wire electrode holder (XYEH01-1; Xuyue Sci. and Tech. Co., Ltd., Beijing, China) was used to make electrical contact with the electrolyte solution. The reference electrode was DRIREF-2 (World Precision Instruments, www.wpiinc.com). Ion-selective electrodes were calibrated prior to flux measurements (The concentration of Na^+^ was usually 0.1 mM in the measuring buffer for root samples). Na^+^ flux was calculated by Fick’s law of diffuseon: *J* = −*D(dc/dx)*, where *J* represents the Na^+^ flux in the *x* direction, *dc/dx* is the Na^+^-concentration gradient, and *D* is the Na^+^ diffusion constant.

### 4.7. H_2_O_2_ Production with Root Cells

A specific fluorescent probe, 2′,7′-dichlorodihydrofluorescein diacetate (H_2_DCF-DA; Molecular Probes) was used for H_2_O_2_ detection in the roots of GL1 plants and *CPK12*-RNAi plants. Shock and short-term responses of H_2_O_2_ to NaCl exposure were examined in this study.

Seven-day-old seedlings (GL1 and *CPK12*-RNAi) grown on MS medium were exposed to 0 or 100 mM NaCl for 10 min, 12 h, and 24 h and then incubated with 10 μM H_2_DCF-DA (prepared in liquid MS medium, pH 5.7) for 5 min at room temperature in the dark. The H_2_DCF-DA-loaded seedlings were washed 3–4 times with liquid MS solution. DCF-specific fluorescence was examined under a Leica SP5 confocal microscope (Leica Microsystem GmbH, Wetzlar, Germany), with excitation at 488 nm and emission at 510–530 nm. Relative H_2_DCF-DA fluorescence intensities in root cells were measured with ImageJ 1.48 (National Institutes of Health, Bethesda, MD, USA).

### 4.8. Activity Analyses of Antioxidant Enzymes

For the detection of antioxidant enzyme activities, after salt treatment for ten days, the *Arabidopsis* seedling (0.2 g) was ground to a fine powder in liquid nitrogen, and then 2 mL ice-cold 50 mM potassium phosphate buffer (pH 7.0) was added. After centrifugation at 12,000 *g* for 20 min, the supernatant was used to detect the enzymatic activities of the antioxidant enzymes SOD, CAT, and APX. The activities of antioxidant enzymes were determined using commercial kits (Nanjing Jiancheng Bioengineering Institute, Nanjing, China). SOD activity was measured using the Superoxide Dismutase WST-1 Assay Kit, which, based on the xanthine/xanthine oxidase method, depended on the production of O^2−^ anions. CAT activity was measured by analyzing the yellowish complex produced by the reaction between H_2_O_2_ and ammonium molybdate and calculating CAT activity by measuring OD value at 405 nm. APX activity was estimated based on the reaction of ASA with H_2_O_2_ to oxidize ASA to MDASA; APX activity was calculated by measuring the reduced OD value at 290 nm. Activities of SOD and CAT are expressed as units per milligrams of protein (U/mg protein). The activity of APX is expressed as (U/g protein).

### 4.9. Cell Viability Analyses

Seven-day-old seedlings (GL1 and *CPK12*-RNAi) grown on MS medium were exposed to 0 or 100 mM NaCl for 12 h, and cell viability was measured by staining seedlings with FDA (Invitrogen, Carlsbad, CA, USA). The confocal parameters were set as described in previous studies: the excitation wavelength was 488 nm, and the emission wavelengths were 505 to 525 nm. Relative FDA fluorescence intensities in root cells were measured with ImageJ 1.48 (National Institutes of Health, Bethesda, MD, USA).

### 4.10. Data Analysis

All experimental data were analyzed with SPSS version 17.0 software (IBM China Company Ltd., Beijing, China) for statistical evaluations. Statistical analysis were performed using one-way ANOVA. Differences were considered significant at *p* < 0.05, unless otherwise stated.

## Figures and Tables

**Figure 1 ijms-19-04062-f001:**
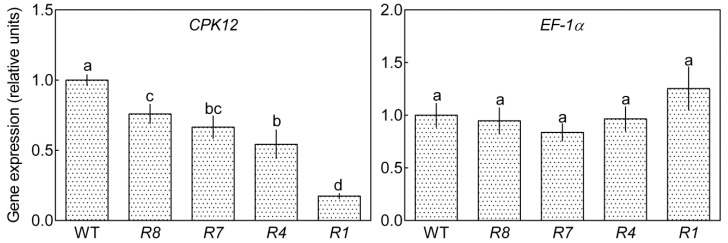
The expression of *CPK12* (*Ca^2+^-Dependent Protein Kinase 12*) and *EF-1α* (*Elongation Factor-1α*) in WT and *CPK12*-RNAi mutant. The mRNA levels (relative units, normalized relative to the mRNA level of the wild-type GL1 taken as 100%) of *CPK12* and *EF-1α*, estimated by qRT-PCR, in the non-transgenic GL1 (WT) and four different transgenic *CPK12*-RNAi lines (indicated by *R8*, *R7*, *R4*, and *R1*). Values are mean ± standard error from three independent experiments. Columns labeled with different letters indicate significant differences at *p* < 0.05.

**Figure 2 ijms-19-04062-f002:**
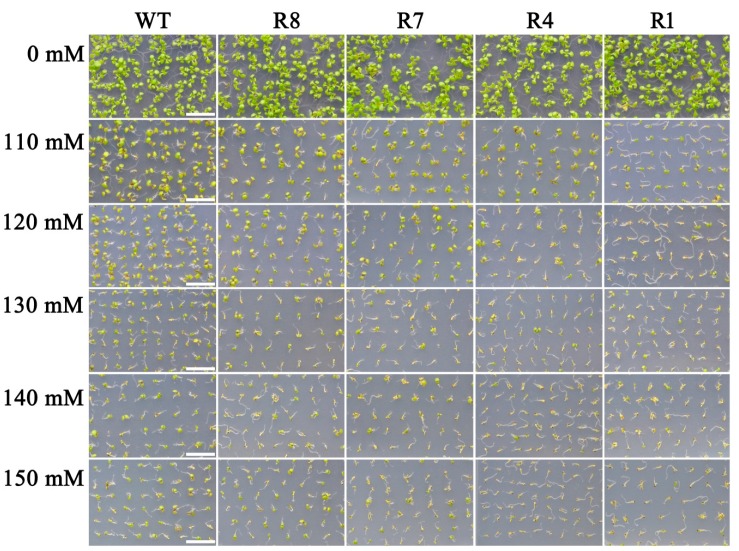
Down-regulation of *CPK12* results in NaCl-hypersensitive seedling growth. Seeds were planted in NaCl-free (0 mM) medium or media containing 110, 120, 130, 140, or 150 mM NaCl, and seedling growth was investigated 10 days after stratification. Scale bars, 1 cm.

**Figure 3 ijms-19-04062-f003:**
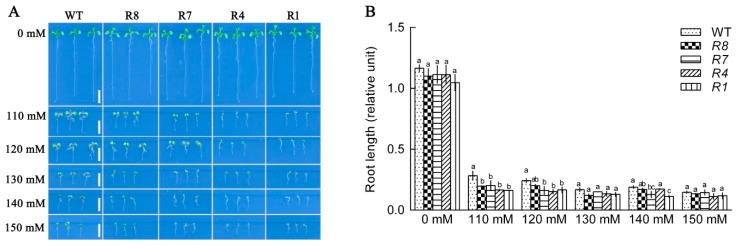
Down-regulation of *CPK12* results in NaCl-hypersensitive root growth. (**A**) Seeds were planted in NaCl-free (0 mM) medium or media containing 110, 120, 130, 140, 150, or mM NaCl, and root growth was detected 10 days after stratification. Scale bars, 0.5 cm. (**B**) Root lengths are mean ± standard error from three independent experiments. Thirty plants were measured for each genotype in each treatment. The mean values of root lengths are labeled with letters in the same group to denote significant differences (*p* < 0.05).

**Figure 4 ijms-19-04062-f004:**
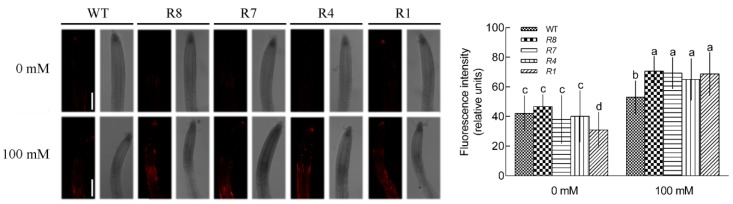
Ca^2+^ levels within roots of GL1 and *CPK12*-RNAi plants. (**A**) Seven-day-old seedlings were transferred to MS medium supplemented with (100 mM) or without NaCl (0 mM) for 12 h, then stained with the Ca^2+^-specific fluorescent probe Rhod-2 AM for 1 h at room temperature. Orange-red fluorescence within cells was detected at the apical region of roots under a Leica confocal microscope. Representative confocal images show cytosolic Ca^2+^ content in plant roots. Scale bars, 100 μm. (**B**) The relative fluorescence intensity (±SD) represents the mean of 10 independent seedlings. The mean values of Ca^2+^ fluorescence are labeled with letters in the same group to denote significant differences (*p* < 0.05).

**Figure 5 ijms-19-04062-f005:**
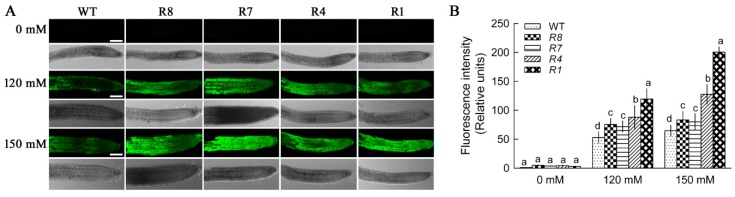
Na^+^ levels in root cells of wild-type (WT) GL1 and *CPK12*-RNAi plants under salt stress. (**A**) Seven-day-old seedlings were transferred to MS medium supplemented with (120 mM, 150 mM) or without NaCl (0 mM) for 12 h, then seedlings were treated with CoroNa-Green AM (green fluorescence, sodium-specific) for 1 h. Green fluorescence in root cells was observed at the apical region of roots using a Leica confocal microscope. Typical images show Na^+^ content in plant roots. Scale bars, 100 μm. (**B**) The mean relative fluorescence values marked with letters in the same group represent significant differences (*p* < 0.05).

**Figure 6 ijms-19-04062-f006:**
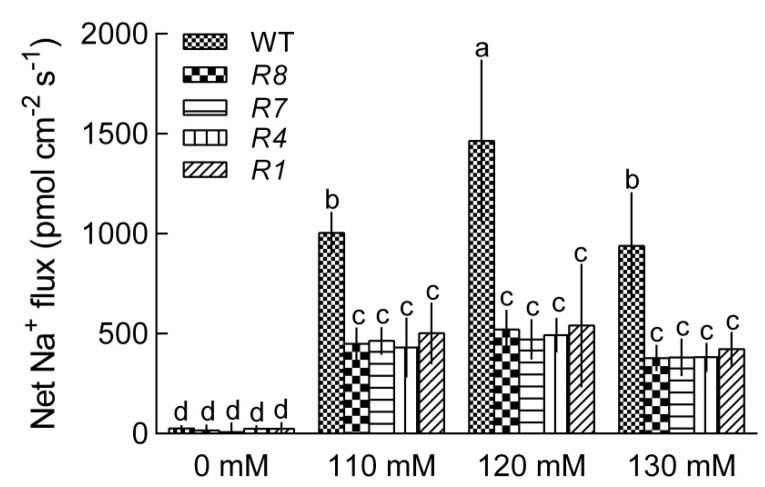
Na^+^ flux in GL1 and *CPK12*-RNAi plants. Seeds were germinated for one week in a vertical direction on MS agar medium containing 0, 110, 120, 130 mM NaCl. Continuous NMT recording were applied at the meristem region of the root tips. Each column is the mean of six independent seedlings; bars show the standard error of the mean. Columns marked with letters in the same group indicate significant differences at *p* < 0.05.

**Figure 7 ijms-19-04062-f007:**
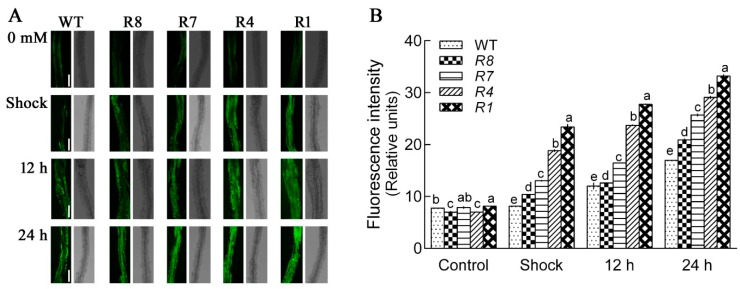
Accumulation of H_2_O_2_ in the root tips of GL1 and *CPK12*-RNAi plants exposed to salt stress. (**A**) After germinating seven days, the *Arabidopsis* seedlings were transferred to MS medium containing 0 or 100 mM NaCl for 10 min, 12 h, or 24 h. These seedlings were incubated with H_2_DCF-DA for 5 min. The green fluorescence within cells at the apical region of roots was detected using a Leica confocal microscope. Scale bars, 100 μm. (**B**) The relative fluorescence intensity (±SD) represents the mean of 10 *Arabidopsis* seedlings. The mean values of H_2_O_2_ fluorescence are labeled with letters in the same group to denote significant differences (*p* < 0.05).

**Figure 8 ijms-19-04062-f008:**
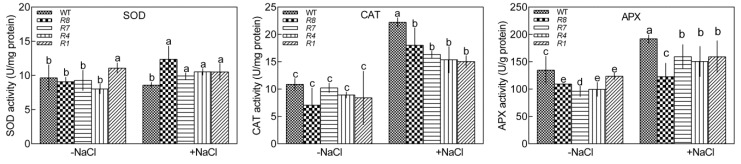
Effect of NaCl on activities of superoxide dismutase (SOD), catalase (CAT), and ascorbate peroxidase (APX) in wild-type (GL1) and *CPK12*-RNAi lines. Seven-day-old seedlings were transferred to MS medium supplemented with or without 100 mM NaCl for 10 d. The activities of antioxidant enzymes were analyzed. Each column shows the mean of three replicated experiments and bars represent the standard error of the mean. Columns labeled with letters in the same group denote significant difference at *p* < 0.05.

**Figure 9 ijms-19-04062-f009:**
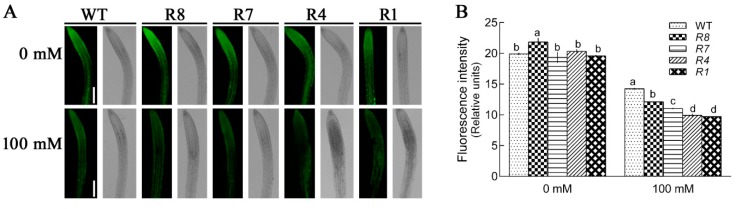
Effect of salt stress on cell viability in wild-type (GL1) and *CPK12*-RNAi lines. (**A**) Seven-day-old seedlings were transferred to MS medium supplemented with or without 100 mM NaCl for 12 h. Cell viability was assayed with fluorescein diacetate (FDA, green) stain. Representative images of apical region of roots are shown. Scale bars, 100 μm. (**B**) The fluorescence intensity (±SD) represents the mean of 10 independent seedlings. The mean values of FDA fluorescence are labeled with letters in the same group to denote significant differences (*p* < 0.05).

**Figure 10 ijms-19-04062-f010:**
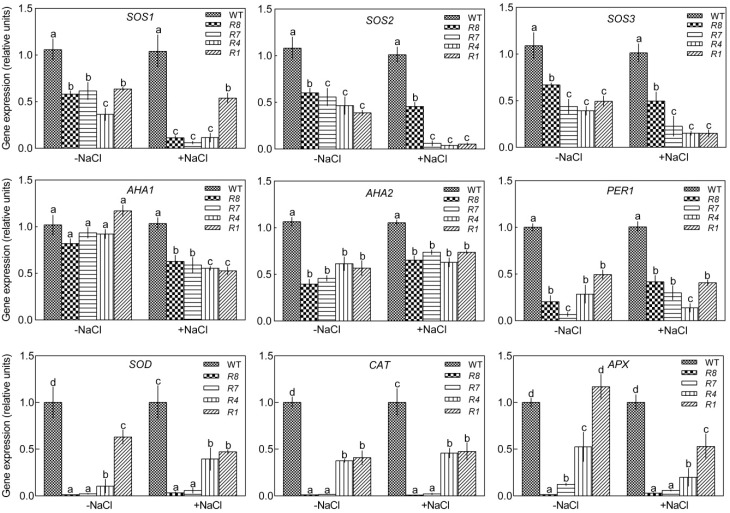
Changes in *CPK12* expression alter expression of a subset of genes involved in salt stress responses. The mRNA levels in the seedling of wild-type GL1, *CPK12*-RNAi mutants were determined by qRT-PCR. One-week-old seedlings were transferred to MS medium with and without the addition of 100 mM NaCl for ten days. The expression of salt stress responsive genes was analyzed. The gene expression levels were normalized relative to the value of the GL1 plants. Each value is the mean of the three independent determinations; columns labeled with letters in the same group indicate significant differences (*p* < 0.05).

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
