# Peer review of "The Arabidopsis Ca2+-Dependent Protein Kinase CPK12 Is Involved in Plant Response to Salt Stress"

_ijms, 2018, doi:10.3390/ijms19124062_

Reviewer 1 Report

The work done by Zhang et al. demonstrates that the Ca2+ dependent protein kinase (CDPK12) is essential for salt tolerance in Arabidopsis. Down regulation of CDPK12 in Arabidopsis results in accumulation of Na+ and H2O2 after salt stress. The Na+ and H2O2 accumulation occurs in a dose dependent manner. Additionally the accumulation of Na+ and H2O2 increases with the decrease in the expression of CDPK12. Plants with down regulated CDPK12 are incapable of removing excess Na+ from the cell. This is evident from the altered expression of the salt responsive genes in the mutants even in the absence of NaCl. Clearly CDPK 12 is found to be an important player in response to salt stress in Arabidopsis. However, I have the following major concerns:

1. The author’s data show that after NaCl treatment the cpk12-RNAi plants accumulate more H2O2 than the control plants. The authors mention that CDPK12 is involved in the elimination of H2O2 under salt stress. However they do not provide any evidence for this explanation. It would be logical to check the expression level of some ROS scavenging genes like Catalase, SOD, APX etc.

2. Do the cpk12-RNAi plants have a general problem in H2O2 elimination or they are incapable of NaCl-induced H2O2 scavenging only?

3. NaCl treatment also induces a Ca2+ elevation; it would be informative to check the Ca2+level in the cpk12-RNAi plants after salt stress.

4. In the cpk12-RNAi plants, except AHA1 all the salt responsive genes tested have lower expression than the wt plants in unstressed condition. The authors must shed some light into this finding.

5. The authors should also check the expression of a control gene (not related to salt-response) in the cpk12-RNAi plants without NaCl treatment.

6.  The authors must improve the discussion part. The results obtained should be discussed in detail in the discussion.

Some minor comments:

The authors mention that the cpk12-RNAi plants display early seedling growth: however, no data is provided to support this. The authors should check the manuscript for some minor typing errors.

Author Response

QUESTION 1:

The author’s data show that after NaCl treatment the cpk12-RNAi plants accumulate more H2O2 than the control plants. The authors mention that CDPK12 is involved in the elimination of H2O2 under salt stress. However they do not provide any evidence for this explanation. It would be logical to check the expression level of some ROS scavenging genes like Catalase, SOD, APX etc.

ANSWER:

We detected the expression of SOD, CAT, and APX genes in GL1 and CPK12-RNAi plants, in the absence of salt treatment, the expression of SOD and CAT was down-regulated in CPK12-RNAi plants, and APX was down-regulated in R8, R7, and R4 plants. Under salt stress, the expression of APX was down-regulated in R1. It is interested that the expression of SOD, CAT, and APX was nearly undetectable in R8 and R7 plants, whether under salt or no-salt stress. These results indicate that CPK12 regulates the expression of SOD, CAT, and APX (Figure 10).

In additionally, we measured the activities of antioxidant enzymes, such as SOD, CAT, and APX, in GL1 and CPK12-RNAi plants. In the absence of salt treatment, the activities of SOD and CAT were not significant between GL1 and CPK12-RNAi plants, but the activity of APX in CPK12-RNAi plants was lower than GL1. Under salt-stress treatment, the activity of SOD in GL1 was higher than CPK12-RNAi plants, but CAT and APX were lower in CPK12-RNAi plants which compared with GL1 (Figure 8).These data imply that CPK12 is involved in the elimination of H2O2 under salt stress.

QUESTION 2:

Do the cpk12-RNAi plants have a general problem in H2O2 elimination or they are incapable of NaCl-induced H2O2 scavenging only?

ANSWER

This is a very good question. Without salt-stress, the level of H2O2, the activity of SOD and CAT were not significant between GL1 and CPK12-RNAi plants (Figure 8). Under NaCl stress, H2O2 levels in the roots of CPK12-RNAi plants increased and were higher than levels in GL1 plants (Figure 7), and the activity of SOD was higher in CPK12-RNAi plants than GL1, but the activity of CAT was lower in CPK12-RNAi plants than GL1 (Figure 8). The enhanced activity of SOD in CPK12-RNAi plants results in H2O2 production, but the reduced activity of CAT cause H2O2 accumulation, these results suggest that down-regulated CPK12 cannot scavenging NaCl-induced H2O2 burst.

QUESTION 3:

NaCl treatment also induces a Ca2+ elevation; it would be informative to check the Ca2+ level in the cpk12-RNAi plants after salt stress.

ANSWER:

We examined the Ca2+ level in the CPK12-RNAi plants and wild type plants GL1 after salt stress using the Ca2+ specific probe, Rhod-2 AM. In the absence of salt treatment, the relative fluorescence intensity was not significantly between GL1 and CPK12-RNAi plants, except the R1 line, probably due to the lowest expression of CPK12 in R1 line. Under salt treatment, Ca2+ levels in the roots of CPK12-RNAi and GL1 plants increased (Figure 4).

QUESTION 4:

In the cpk12-RNAi plants, except AHA1 all the salt responsive genes tested have lower expression than the wt plants in unstressed condition. The authors must shed some light into this finding.

ANSWER:

In this work, we examined the expression of some salt related gene. Compared with GL1, except AHA1, all the salt responsive genes were down-regulated in CPK12-RNAi plants in unstressed condition (Figure 10). It is interested that down-regulated CPK12 was not influence the post-germination seedling growth (Figure 2, 3), and the expression of a control gene EF-1α, which is not related to salt-stress (Figure 1), additionally, the activity of SOD and CAT were not affected in CPK12-RNAi plants without salt stress treatment (Figure 8), indicating that CPK12 is involved in salt stress signaling transduction, but not relate to seeding development.

QUESTION 5:

The authors should also check the expression of a control gene (not related to salt-response) in the cpk12-RNAi plants without NaCl treatment.

ANSWER:

We examined the expression of a control gene EF-1α, which is not related to salt-stress, was not affected in those CPK12-RNAi lines (Figure 1), indicating that CPK12 is involved in stress signaling transduction, but not relate to seedling development.

QUESTION 6:

The authors must improve the discussion part. The results obtained should be discussed in detail in the discussion.

ANSWER:

Thank you very much for your suggestion. We have rewritten the discussion part, analyzed and discussed in detail in the discussion.

QUESTION 7:

The authors mention that the cpk12-RNAi plants display early seedling growth: however, no data is provided to support this. The authors should check the manuscript for some minor typing errors.

ANSWER:

I’m sorry that this is a typing error and we changed the statement: “CPK12-RNAi mutants exhibited the same post-germination seedling growth status with GL1 plants in the free of NaCl medium”.

We have checked the manuscript carefully and correct minor typing errors.

Reviewer 2 Report

The authors of «The Arabidopsis Ca2+-dependent protein kinase CPK12 is involved in plant response to salt stress » present new data on the role of CPK12 in salt stress tolerance in planta using their RNAi transgenic lines already established from their previous work. The manuscript is well written, clear in its presentation and the results are convincing. Nevertheless, before this work could be accepted for publication I have some recommendations concerning the quantitation of Na+ using Coro-Na-Green probe.

1) Indeed, this probe have been developed for very precise imaging purposes. This probe is not suitable for fluorescence lifetime imaging microscopy measurements because of this absence of Na-dependent changes in the exponential decay time. Its relatively high sodium dissociation constant of 82 mM, make it well suitable for the measurement of very large Na+ transients changes or Na+ changes upon a high background Na+ concentration. I guess that depending on the Na+ efflux dynamic, some information could be missed. Moreover the selectivity of this probe Na+ versus K+ binding is only about 4-fold, thus considerably lower than that of Sodium Green probe. The authors have to include a discussion about this last point and I suggest presenting the quantitative results as a “relative” quantitation of Na+.

2) Minor changes:

- I think the correct abbreviation for GL1 ecotype is GL1 since gl1 is also used for the glabra1 mutant. Please verify and correct if necessary.

- introduction l37: adaptation is miss spelled

- Please specify in figure legends that statistical analysis were performed for each treatment independently.

- p8, l16: the sentence is not clear. In my opinion, there is an absence of phosphorylation of RBOHB in this quadruple mutant that led to a decrease ROS production. Is it correct?

Author Response

QUESTION 1:

Indeed, this probe (Coro-Na-Green) have been developed for very precise imaging purposes. This probe is not suitable for fluorescence lifetime imaging microscopy measurements because of this absence of Na-dependent changes in the exponential decay time. Its relatively high sodium dissociation constant of 82 mM, make it well suitable for the measurement of very large Na+ transients changes or Na+ changes upon a high background Na+ concentration. I guess that depending on the Na+ efflux dynamic, some information could be missed. Moreover the selectivity of this probe Na+ versus K+ binding is only about 4-fold, thus considerably lower than that of Sodium Green probe. The authors have to include a discussion about this last point and I suggest presenting the quantitative results as a “relative” quantitation of Na+.

ANSWER:

Thank you very much and this is a very good question about Na+ probe. There are several commercial Na+ specific probe, for example, SBFI, Sodium Green, CoroNa etc., Sodium Green displays a modest fluorescence increase in response to Na+ binding [1,2], while CoroNa is more suitable for detecting Na+ in a wider range of concentrations, and the selectivity of CoroNa is 4 times higher to Na+ than to K+ binding [1], but CoroNa is not suitable for the detection of relatively low Na+ changes in cells [2]. In this work, after NaCl treatment, the roots absorbed and accumulated high level of Na+, and the Na+ efflux was inhibited. Our previous studies showed that CoroNa is suitable for detecting Na+ level after NaCl treatment in tobacco, Arabidopsis, and Glycyrrhiza uralensis [3,4,5], so we selected CoroNa Green to detect the cytosolic Na+ level in this work.

1. Martin, V.V.; Rothe, A.; Gee, K.R. Fluorescent metal ion indicators based on benzoannelated crown systems: a green fluorescent indicator for intracellular sodium ions. Bioorg. Med. Chem. Lett. 2005, 15, 1851–1855.

2. Iamshanova, O.; Mariot, P.; Lehen'kyi, V.; Prevarskaya, N. Comparison of fluorescence probes for intracellular sodium imaging in prostate cancer cell lines. Eur. Biophys. J. 2016, 45, 765–777.

3. Han, Y.; Wang, W.; Sun, J.; Ding, M.; Zhao, R.; Deng, S.; Wang, F.; Hu, Y.; Wang, Y.; Lu, Y.; Du, L.; Hu, Z.; Diekmann, H.; Shen, X.; Polle, A.; Chen, S. Populus euphratica XTH overexpression enhances salinity tolerance by the development of leaf succulence in transgenic tobacco plants. J. Exp. Bot. 2013, 64, 4225–4238.

4. Zhang, Y.N.; Wang, Y.; Sa, G.; Zhang, Y.H.; Deng, J.Y.; Deng, S.R.; Wang, M.J.; Zhang H,L.; Yao, J.; Ma, X.Y.; Zhao, R.; Zhou, X.Y.; Lu, C.F.; Lin, S.Z.; Chen, S.L. Populus euphratica J3 mediates root K+/Na+ homeostasis by activating plasma membrane H+-ATPase in transgenic Arabidopsis under NaCl salinity. Plant Cell Tiss. Organ Cult. 2017, 131, 75–88.

5. Lang, T.; Deng, S.; Zhao, N.; Deng, C.; Zhang, Y.; Zhang, Y.; Zhang, H.; Sa, G.; Yao, J.; Wu, C.; Wu, Y.; Deng, Q.; Lin, S.; Xia, J.; Chen, S. Salt-sensitive signaling networks in the mediation of K+/Na+ homeostasis gene expression in Glycyrrhiza uralensis roots. Front. Plant Sci. 2017, 8, 1403.

QUESTION 2:

I think the correct abbreviation for GL1 ecotype is GL1 since gl1 is also used for the glabra1 mutant. Please verify and correct if necessary.

ANSWER:

This is a good suggestion, and we changed gl1 to GL1.

QUESTION 3:

introduction l37: adaptation is miss spelled

ANSWER:

I apologize for our negligence. We corrected the typing errors about adaptation.

QUESTION 4:

Please specify in figure legends that statistical analysis were performed for each treatment independently.

ANSWER:

The statistical analyses were performed using one-way ANOVA for experiments.

QUESTION 5:

p8, l16: the sentence is not clear. In my opinion, there is an absence of phosphorylation of RBOHB in this quadruple mutant that led to a decrease ROS production. Is it correct?

ANSWER:

We modified the sentence, “In contrast, Arabidopsis cpk5 cpk6 cpk11 cpk4 quadruple mutants harbor decreased ROS content, suggesting that these CDPKs regulate ROS production potentially by phosphorylating the NADPH oxidase RBOHB.”

Round  2

Reviewer 1 Report

The authors have answered to all my comments.

This revised work can be accepted in its present form. 

Author Response

Comment 1: The authors have answered to all my comments. This revised work can be accepted in its present form.

Reply: Thank you very much for your positive comments.

Reviewer 2 Report

The authors have addresses all the points. However, still there are several grammatical errors in the manuscript. I am pointing out some below

Page 5- line 45: positive regulate should be positively regulate

            Line 49: please clarify what is ACPK1

Page 7- line 2:  significant between should be significantly different between

            Line 4: under salt stress condition/ upon salt stress treatment. Please reformulate the entire sentence this sentence is not clear

Page 8-line 7: It is interesting instead of it is interested

            Line15: gene expression levels were “in”

            Line 22: signaling transduction should be signal transduction, implying should be imply

            Line 24: gene was should be genes were

These are just some examples there are more errors in the manuscript. Kindly go through the entire manuscript carefully and correct these kinds of errors.

Author Response

Comment 1: Page 5- line 45: positive regulate should be positively regulate

Reply: We have changed positive to positively.

Comment 2: Line 49: please clarify what is ACPK1

Reply: ACPK1 means ABA-stimulated calcium-dependent protein kinase1. Please see the reference, Yu, X.C.; Li, M.J.; Gao, G.F.; Feng, H.Z.; Geng, X.Q.; Peng, C.C.; Zhu, S.Y.; Wang, X.J.; Shen, Y.Y.; Zhang, D.P. Abscisic acid stimulates a calcium-dependent protein kinase in grape berry. Plant Physiol. 2006, 140, 558–579.

In the abstract, there is a sentence, Here we identified, characterized, and purified a 58-kD ABA-stimulated calcium-dependent protein kinase from the mesocarp of grape berries (Vitis vinifera×Vitis labrusca), designated ACPK1 (for ABA-stimulated calcium-dependent protein kinase1).

 Comment 3: Page 7- line 2: significant between should be significantly different between

Reply: We have changed “significant” to “significantly”.

Comment 4: Line 4: under salt stress condition/ upon salt stress treatment. Please reformulate the entire sentence this sentence is not clear

Reply: We have changed the sentence to “under salt stress condition”.

Comment 5: Page 8-line 7: It is interesting instead of it is interested

Reply: We have changed “It is interested” to “It is interesting”.

Comment 6: Line15: gene expression levels were “in”

Reply: We have added “in” behind “The gene expression levels were”.

Comment 7: Line 22: signaling transduction should be signal transduction, implying should be imply

Reply: We have changed “signaling transduction” to “signal transduction”, “implying” to “imply”.

Comment 8: Line 24: gene was should be genes were

Reply: We have changed “gene was” to “genes were”.

Comment 9: These are just some examples there are more errors in the manuscript. Kindly go through the entire manuscript carefully and correct these kinds of errors.

Reply: We have checked the entire manuscript and corrected the grammatical errors.